# Volatility Prediction using Financial Disclosures Sentiments with Word Embedding-based IR Models

## Abstract

Volatility prediction—an essential concept in financial markets—has recently been addressed using sentiment analysis methods. We investigate the sentiment of annual disclosures of companies in stock markets to forecast volatility. We specifically explore the use of recent Information Retrieval (IR) term weighting models that are effectively extended by related terms using word embeddings. In parallel to textual information, factual market data have been widely used as the mainstream approach to forecast market risk. We therefore study different fusion methods to combine text and market data resources. Our word embedding-based approach significantly outperforms state-of-the-art methods. In addition, we investigate the characteristics of the reports of the companies in different financial sectors.

## 1 Introduction

Financial volatility is an essential indicator of instability and risk of a company, sector or economy. Volatility forecasting has gained considerable attention during the last three decades. In addition to using historic stock prices, new methods in this domain use sentiment analysis to exploit various text resources, such as financial reports (Kogan et al., 2009; Wang et al., 2013; Tsai and Wang, 2014; Nopp and Hanbury, 2015), news (Kazemian et al., 2014; Ding et al., 2015), message boards (Nguyen and Shirai, 2015), and earning calls (Wang and Hua, 2014).

An interesting resource of textual information are the companies' annual disclosures, known as *10-K filing* reports. They contain comprehensive information about the companies' business as well as risk factors. Specifically, section *Item 1A - Risk Factors* of the reports contains information about the most significant risks for the company. These reports are however long, redundant, and written in a style that makes them complex to process. Dyer et al. (2016) notes that: *"10-K reports are getting more redundant and complex [...] (it) requires a reader to have 21.6 years of formal education to fully comprehend"*. Dyer et al. also analyse the topics discussed in the reports and observe a constant increase over the years in both the length of the documents as well as the number of topics. They claim that the increase in length is not the result of economic factors but is due to verboseness and redundancy in the reports. They suggest that only the risk factors topic appears to be useful and informative to investors. Their analysis motivates us to study the effectiveness of the Risk Factors section for volatility prediction.

Our research builds on previous studies on volatility prediction and information analysis of 10-K reports using sentiment analysis (Kogan et al., 2009; Tsai and Wang, 2014; Wang et al., 2013; Nopp and Hanbury, 2015; Li, 2010; Campbell et al., 2014), in the sense that since the reports are long (average length of 5000 words), different approaches are required, compared with studies of sentiment analysis on short-texts. Such previous studies on 10-K reports have mostly used the data before 2008 and there is little work on the analysis of the informativeness and effectiveness of the recent reports with regards to volatility prediction. We will indeed show that the content of the reports changes significantly not only before and after 2008, but rather in a cycle of 3-4 years.

In terms of use of the textual content for volatility prediction, this paper shows that state-of-the-art Information Retrieval (IR) term weighting models, which benefit from word embedding information, have a significantly positive impact on prediction accuracy. The most recent study on

the topic (Tsai and Wang, 2014) used related terms obtained by word embeddings to expand the lexicon of sentiment terms. In contrast, similar to Rekabsaz et al. (2016b), we define the weight of each lexicon term by extending it to the similar terms in the document. The significant improvement of this approach for document retrieval by capturing the importance of the terms motivates us to apply it on sentiment analysis. We extensively evaluate various state-of-the-art sentiment analysis methods to investigate the effectiveness of our approach.

In addition to text, factual market data (i.e. historical prices) provide valuable resources for volatility prediction e.g. in the framework of GARCH models (Engle, 1982). An emerging question is how to approach the combination of the textual and factual market information. We propose various methods for this issue and show the performance and characteristics of each.

The financial system covers a wide variety of industries, from daily-consumption products to space mission technologies. It is intuitive to consider that the factors of instability and uncertainty are different between the various sectors while similar inside them. We therefore also analyse the sentiment of the reports of each sector separately and study their particular characteristics.

The present study shows the value of information in the 10-K reports for volatility prediction. Our proposed approach to sentiment analysis significantly outperforms state-of-the-art methods (Kogan et al., 2009; Tsai and Wang, 2014; Wang et al., 2013). We also show that performance can be further improved by effectively combining textual and factual market information. In addition, we shed light on the effects of tailoring the analysis to each sector: despite the reasonable expectation that domain-specific training would lead to improvements, we show that our general model generalizes well and outperforms sector-specific trained models.

The remainder of the paper is organized as follows: in the next section, we review the state-of-the-art and related studies. Section 3 formulates the problem, followed by a detailed explanation of our approach in Section 4. We explain the dataset and settings of the experiments in Section 5, followed by the full description of the experiments in Section 6. We conclude the work in Section 7.

## 2 Related Work

Market prediction has been attracting much attention in recent years in the natural language processing community. Kazemian et al. (2014) use sentiment analysis for predicting stock price movements in a simulated security trading system using news data, showing the advantages of the method against simple trading strategies. Ding et al. (2015) address a similar objective while using deep learning to extract and learn events in the news. Xie et al. (2013) introduce a semantic tree-based model to represent news data for predicting stock price movement. Luss et al. (2015) also exploit news in combination with return prices to predict intra-day price movements. They use the Multi Kernel Learning (MKL) algorithm for combining the two features. The combination shows improvement in final prediction in comparison to using each of the features alone. Motivated by this study, we investigate the performance of the MKL algorithm as one of the methods to combine the textual with non-textual information. Other data resources, such as stocks' message boards, are used by Nguyen and Shirai (2015) to study topic modelling for aspect-based sentiment analysis. Wang and Hua (2014) investigate the sentiment of the transcript of earning calls for volatility prediction using the Gaussian Copula regression model.

While the mentioned studies use short-length texts (sentence or paragraph level), approaching long texts (document level) for market prediction is mainly based on n-gram bag of words methods. Nopp and Hanbury (2015) study the sentiment of banks' annual reports to assess banking systems risk factors using a finance-specific lexicon, provided by Loughran and McDonald (2011), in both unsupervised and supervised manner.

More directly related to the informativeness of the 10-K reports for volatility prediction, Kogan et al. (2009) use a linear Support Vector Machine (SVM) algorithm on the reports published between 1996–2006. Wang et al. (2013) improve upon this by using the Loughran and McDonald (2011) lexicon, observing improvement in the prediction. Later, Tsai and Wang (2014) apply the same method as Wang et al. (2013) while additionally using word embedding to expand the financial lexicon. We reproduce all the methods in these studies, and show the advantage of our sentiment analysis approach.

## 3 Problem Formulation

In this section, we formulate the volatility forecasting problem and the prediction objectives of our experiments. Similar to previous studies (Christiansen et al., 2012; Kogan et al., 2009; Tsai and Wang, 2014), volatility is defined as the natural log of the standard deviation of (adjusted) return prices in a window of $\tau$ days. This definition is referred to as standard volatility (Li and Hong, 2011) or realized volatility (Liu and TSE, 2013), defined as follows:

$$v_{[s,s+\tau]} = ln\left(\sqrt{\frac{\sum_{t=s}^{s+\tau}(r_t - \bar{r})^2}{\tau}}\right) \quad (1)$$

where $r_t$ is the return price and $\bar{r}$ the mean of return prices. The return price is calculated by $r_t = ln(P_t) - ln(P_{t-1})$, where $P_t$ is the (adjusted) closing price of a given stock at the trading date $t$.

Given an arbitrary report $i$, we define a prediction label $y_i^k$ as the volatility of the stock of the reporting company in the $k$th quarter-sized window starting from the issue date of the report $s_i$:

$$y_i^k = v_{[s_i+64(k-1),s_i+64k]} \quad (2)$$

Every quarter is considered as per convention, 64 working days, while the full year is assumed to have 256 working days.

We use 8 learners for labels $y^1$ to $y^8$. For brevity, unless otherwise mentioned, we report the volatility of the first year by calculating the mean of the first four quartiles after the publication of each report.

## 4 Methodology

We first describe our text sentiment analysis methods, followed by the features obtained from factual market data, and finally explain the methods to combine textual and market feature sets.

### 4.1 Sentiment Analysis

Similar to previous studies (Nopp and Hanbury, 2015; Wang et al., 2013), we extract the keyword set from a finance-specific lexicon (Loughran and McDonald, 2011) using the positive, negative, and uncertain groups, stemmed using the Porter stemmer. We refer to this keyword set as Lex. Tsai and Wang (2014) expanded this set by adding the top 20 related terms to each term to the original set. The related terms are obtained using the Word2Vec (Mikolov et al., 2013) model, built on the corpus of all the reports, with Cosine similarity. We also use this expanded set in our experiments and refer to it as LexExt.

The following word weighting schemes are commonly used in Information Retrieval and we consider them as well in our study:

**TC** : $\quad log(1 + tc_{d_i}(t))$

**TF** : $\quad \frac{log(1+tc_{d_i}(t))}{\|d_i\|}$

**TFIDF** : $\quad \frac{log(1+tc_{d_i}(t))}{\|d_i\|}log(1 + \frac{|d_i|}{df(t)})$

**BM25** : $\frac{(k+1)\overline{tf_{d_i}}(t)}{k+\overline{tf_{d_i}}(t)}, \quad \overline{tf_{d_i}}(t) = \frac{tc_{d_i}(t)}{(1-b)+b\frac{|d_i|}{avgdl}}$

where $tc_{d_i}(t)$ is the number of occurrences of keyword $t$ in report $i$, $\|d_i\|$ denotes the Euclidean norm of the keyword weights of the report, $|d_i|$ is the length of the report (number of the words in the report), $avgdl$ is the average document length, and finally $k$ and $b$ are parameters. For them, we use the settings used in previous studies (Rekabsaz et al., 2016b) i.e. $k = 1.2$ and $b = 0.65$.

In addition to the standard weighting schemes, we use state-of-the-art weighting methods in Information Retrieval (Rekabsaz et al., 2016b) which benefit directly from word embedding models: They exploit similarity values between words provided by the word embedding model into the weighting schemes by extending the weight of each lexicon keyword with its similar words:

$$\widehat{tc_{d_i}}(t) = tc_{d_i}(t) + \sum_{t' \in R(t)} sim(t,t')tc_{d_i}(t') \quad (3)$$

where $R(t)$ is the list of similar words to the keyword $t$, and $sim(t,t')$ is the Cosine similarity value between the vector representations of the words $t$ and $t'$. As previously suggested by Rekabsaz et al. (2016a, 2017), we use the Cosine similarity function with threshold 0.70 for selecting the set $R(t)$ of similar words.

We define the extended versions of the standard weighting schemes as $\widehat{TC}$, $\widehat{TF}$, $\widehat{TFIDF}$, and $\widehat{BM25}$ by replacing $tc_{d_i}(t)$ with $\widehat{tc_{d_i}}(t)$ in each of the schemes.

The feature vector generated by the weights of the Lex or LexExt lexicons is highly sparse, as the number of dimensions is larger than the number of data-points. We therefore reduce the dimensions by applying Principle Component Analysis (PCA). Our initial experiments show 400 dimen-

sion as the optimum by trying on a range of dimensions from 50 to 1000.

Given the final feature vector $x$ with $l$ dimensions, we apply SVM as a well-known method for training both regression and classification methods. Support Vector Regression (Drucker et al., 1997) formulates the training as the following optimization problem:

$$\min_{w \in \mathbb{R}^l} \frac{1}{2} \|w\|^2 + \frac{C}{N} \sum_{i=1}^{N} \max(0, \|y_i - f(x_i; w)\| - \epsilon)$$
(4)

Similar to previous studies (Tsai and Wang, 2014; Kogan et al., 2009), we set $C = 1.0$ and $\epsilon = 0.1$. To solve the above problem, the function $f$ can be re-parametrized in terms of a kernel function $K$ with weights $\alpha_i$:

$$f(x_i; w) = \sum_{i=1}^{N} \alpha_i K(x_i, x)$$
(5)

The kernel can be considered as a (similarity) function between the feature vector of the document and vectors of all the other documents. Our initial experiments showed better performance of the Radial Basis Function (RBF) kernel in comparison to linear and cosine kernels and is therefore used in this paper.

## 4.2 Market Features

In addition to textual features, we define three features using the factual market data and historical prices—referred to as *market features*—as follows:

**Current Volatility** is calculated on the window of one quartile before the issue date of the report: $v_{[s_i-64, s_i]}$.

**GARCH** (Bollerslev, 1986) is a common econometric time-series model used for predicting stock price volatility. Due to lack of space, the implementation details are moved to supplementary materials.

**Sector** is the sector that the corresponding company of the report belongs to, namely energy (ene), basic industries (ind), finance (fin), technology (tech), miscellaneous (misc), consumer nondurables (n-dur), consumer durables (dur), capital goods (capt), consumer services (serv), public utilities (pub), and health care (hlth)[1]. The feature is converted to numerical representation using one-hot encoding.

---

[1] We follow NASDAQ categorization of sectors.

## 4.3 Feature Fusion

To combine the text and market feature sets, the first approach, used also in previous studies ((Kogan et al., 2009; Wang et al., 2013)) is simply joining all the features in one feature space. In the context of multi-model learning, the method is referred to as *early fusion*.

In contrast, *late fusion* approaches first learn a model on each feature set and then use/learn a meta model to combine their results. As our second approach, we use *stacking* (Wolpert, 1992), a special case of late fusion. In stacking, we first split the training set into two parts (70%-30% portions). Using the first portion, we train separate machine learning models for each of the text and market feature sets. Next, we predict labels of the second portion with the trained models and finally train another model to capture the combinations between the outputs of the base models. In our experiments, the final model is always trained with SVM with RBF kernel.

Stacking is computationally inexpensive. However, due to the split of the training set, the base models or the meta model may suffer from lack of training data. A potential approach to learn both the feature sets in one model is the MKL method.

The MKL algorithm (also called *intermediate fusion* (Noble et al., 2004)) extends the kernel of the SVM model by learning (simultaneous to the parameter learning) an optimum combination of several kernels. The MKL problem as formulated in Lanckriet et al. (2004) add the following criterion to Eq. 5 for kernel learning:

$$K^* = \sum_i d_i K_i \quad \text{where} \sum_i d_i = 1, \ d_i \geq 0 \quad (6)$$

where $K_i$ is a predefined kernel. Gönen and Alpaydın (2011) mention two uses of MKL: learning the optimum kernel in SVM, and combining multiple modalities (feature sets) via each kernel.

However, the optimization can be computationally challenging. We use the mklaren method (Stražar and Curk, 2016) which has linear complexity in the number of data instances and kernels. It has been shown to outperform recent multi kernel approximation approaches. We use RBF kernels for both the text and market feature sets.

## 5 Experiment Setup

In this section, we first describe the data, followed by introducing the baselines. We report the parameters applied in various algorithms and describe the evaluation metrics.

**Dataset** We download the reports of companies of the U.S. stock markets from 2006 to 2015 from the U.S. Securities and Exchange Commission (SEC) website[2]. We remove HTML tags and extract the text parts. We extract the Risk Factors section using term matching heuristics. Finally, the texts are stemmed using the Porter stemmer. We calculate the volatility values (Eq 1) and the volatility of the GARCH model based on the stock prices, collected from the Yahoo website. We filter the volatility values greater/smaller than the mean plus/minus three times the standard deviation of all the volatility values[3].

**Baselines GARCH:** although the GARCH model is of market factual information, we use it as a baseline to compare the effectiveness of text-based methods with mainstream approaches.

**Market:** uses all the market features. For both the GARCH and Market baselines, we use an SVM learner with RBF kernel.

**Wang et al. (2013):** they use the `Lex` keyword set with $TC$ weighting scheme and the SVM method. They combine the textual features with current volatility using the early fusion method.

**Tsai et al. (2014):** similar to Wang et al. (2013), while they use the `LexExt` keyword set.

**Evaluation Metrics** As a common metric in volatility prediction, we use the $r^2$ metric (square of the correlation coefficient) for evaluation:

$$r^2 = \left( \frac{\sum_{i=1}^{n} (\hat{y}_i - \bar{\hat{y}})(y_i - \bar{y})}{\sqrt{\sum_{i=1}^{n} (\hat{y}_i - \bar{\hat{y}})^2} \sqrt{\sum_{i=1}^{n} (y_i - \bar{y})^2}} \right)^2 \tag{7}$$

where $\hat{y}_i$ is the predicted value, $y_i$ denotes the labels and $\bar{y}$, their mean. The $r^2$ metric indicates the proportion of variance in the labels explained by the prediction. The measure is close to 1 when the predicted values can explain a large proportion of the variability in the labels and 0 when it fails to explain the labels' variabilities. An alternative metric, used in previous studies (Wang et al.,

---

[2] https://www.sec.gov
[3] The complete dataset is available in anonymousurl

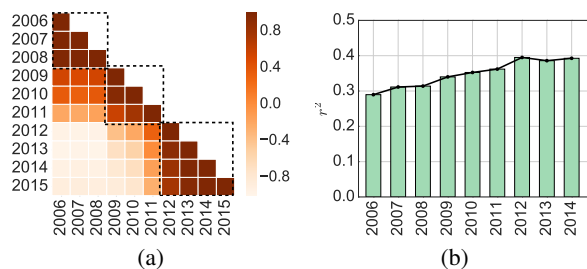

(a) (b)

Figure 1: (a) Cosine similarity between the centroid vectors of the years. (b) Volatility prediction performance when using reports from the specified year to 2015

2013; Tsai and Wang, 2014; Kogan et al., 2009) is Mean Squared Error $MSE = \sum_i (\hat{y}_i - y_i)^2/n$. However, especially when comparing models, applied on different test sets (e.g. performance of first quartile with second quartile), $r^2$ has better interpretability since it is independent of the scale of $y$. We use $r^2$ in all the experiments while the MSE measure is reported only when the models are evaluated on the same test set.

## 6 Experiments and Results

In this section, first we analyse the contents of the reports, followed by studying our sentiment analysis methods for volatility prediction. Finally, we investigate the effect of sentiment analysis of the reports in different industry sectors.

### 6.1 Content Analysis of 10-K Reports

Let us start our experiment with observing changes in the feature vectors of the reports over the years. To compare them, we use the state-of-the-art sentiment analysis method, introduced by Tsai and Wang (2014). We first represent the feature vector of each year by calculating the centroid (element-wise mean) of the feature vectors of all reports published that year and then calculate the Cosine similarity of each pair of centroid vectors, for the years 2006–2015.

Figure 1a shows the similarity heat-map for each pair of the years. We observe a high similarity between three ranges of years: 2006–2008, 2009–2011, and 2012–2015. These considerable differences between the centroid reports in years across these three groups hints at probable issues when using the data of the older years for the more recent ones.

To validate this, we apply 5-fold cross valida-

Table 1: Performance of sentiment analysis methods for the first year.

| Component | Method | Text $(r^2)$ | Text (MSE) | Text+Market $(r^2)$ | Text+Market (MSE) |
|---|---|---|---|---|---|
| Weighting Schema (+Stacking) | $\widehat{BM25}$ | **0.439** | **0.132** | **0.527** | **0.111** |
| | $BM25$ | 0.433 | 0.136 | 0.523 | 0.114 |
| | $\widehat{TC}$ | 0.427 | 0.136 | 0.517 | 0.115 |
| | $TC$ | 0.425 | 0.137 | 0.521 | 0.114 |
| | $\widehat{TFIDF}$ | 0.301 | 0.166 | 0.502 | 0.118 |
| | $TFIDF$ | 0.264 | 0.189 | 0.497 | 0.119 |
| | $\widehat{TF}$ | 0.218 | 0.190 | 0.495 | 0.120 |
| | $TF$ | 0.233 | 0.200 | 0.495 | 0.120 |
| Feature Fusion (+$\widehat{BM25}$) | **Stacking** | - | - | **0.527** | **0.111** |
| | MKL | - | - | 0.488 | 0.126 |
| | Early Fusion | - | - | 0.473 | 0.125 |

Table 2: Performance of the methods using 5-fold cross validation.

| | Method | $(r^2)$ | (MSE) |
|---|---|---|---|
| | GARCH | 0.280 | 0.170 |
| Text | Wang (2013) | 0.345 | 0.154 |
| | Tsai (2014) | 0.395 | 0.142 |
| | Our method | **0.439** | **0.132** |
| | Market | 0.485 | 0.122 |
| Text+Market | Wang (2013) | 0.499 | 0.118 |
| | Tsai (2014) | 0.484 | 0.122 |
| | Our method | **0.527** | **0.111** |

tion, first on all the data (2006–2015), and then on smaller sets by dropping the oldest year i.e. the next subsets use the reports 2007–2015, 2008–2015 and so forth. The results of the $r^2$ measure are shown in Figure 1b. We observe that by dropping the oldest years one by one (from left to right in the figure), the performance starts improving. We argue that this improvement is due to the reduction of noise in data, noise caused by conceptual drifts in the reports as also mentioned by Dyer et al. (2016). In fact, although in machine learning in general using more data results in better generalization of the model and therefore better prediction, the reports of the older years introduce noise.

As shown, the most coherent and largest data consists of the subset of the reports published between 2012 to 2015. This subset is also the most recent cluster and presumably more similar to the future reports. Therefore, in the following, we only use this subset, which consists of 3892 reports, belonging to 1323 companies.

## 6.2 Volatility Prediction

Given the dataset of the 2012–2015 reports, we try all combinations of different term weighting schemes using the `LexExt` keyword set. All weighting schemes are then combined with the market features with the introduced fusion methods. The prediction is done with 5-fold cross validation. The averages of the results of the first four quartiles (first year) are reported in Table 1. To make showing the results tractable, we use the best fusion (stacking) for the weighting schemes and the best scheme ($\widehat{BM25}$) for fusions.

Regarding the weighting schemes, $\widehat{BM25}$, $BM25$, and $\widehat{TC}$ show the best results. In general, the extended schemes (with hat) improve upon their normal forms. For the feature fusion meth-

ods, stacking outperforms the other approaches in both evaluation measures. MKL however has better performance than early fusion while it has the highest computational complexity among the methods. Based on these results, as our best performing approach in the remainder of the paper, we use $\widehat{BM25}$ (with `LexExt` set), reduced to 400 dimensions and stacking as the fusion method. Table 2 summarizes the results of our best performing method compared with previously existing methods. Our method outperforms all state-of-the-art methods both when using textual features only as well as a combination of textual and market features.

Let us now take a closer look on the changes in the performance of the prediction in time. The results of 5-fold cross validation for both tasks on the dataset of the reports, published between 2012–2015 are shown in Figure 2a. The X-axes show eight quartiles after the publication date of the report. For comparison, the GARCH and only market features are depicted with dashed lines.

As shown, the performance of the GARCH method as well as that using only market features (Market) decrease faster in the later quartiles since the historical prices used for prediction become less relevant as time goes by. Using only text features (Text), we see a roughly similar performance between the first four quartiles (first year), while the performance, in general, slightly decreases in the second year. By combining the textual and market features (Text+Market), we see a consistent improvement in comparison to each of them alone. In comparison to using only market features, the combination of the features shows more stable results in the later quartiles. These results support the informativeness of the 10-K reports to more effectively foreseen volatility in long-term

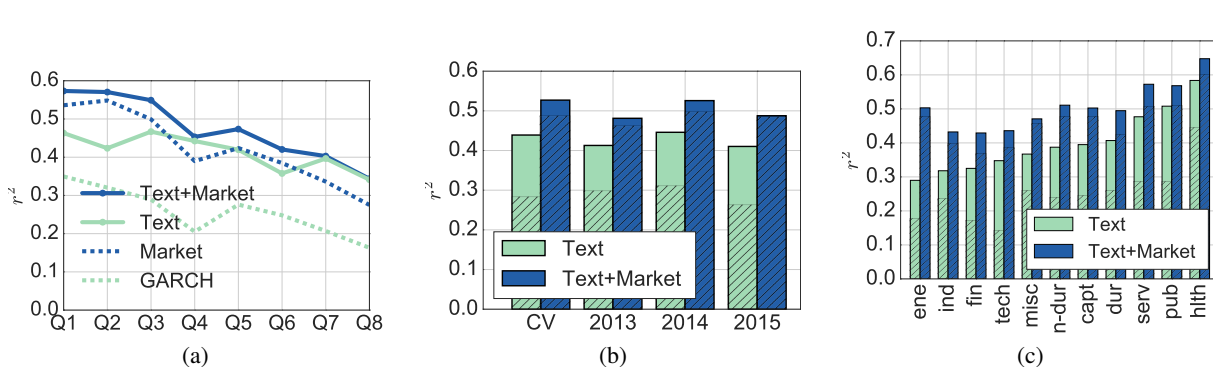

Figure 2: (a) Performance of our approach on 8 quartiles using the Text and Text+Market feature sets. The dashed lines show the market-based baselines. (b) Performance of volatility prediction of each year given the past data. The hashed areas show corresponding baselines. (c) Performance per sector. Abbreviations are defined in Section 4.2

windows.

While the above experiments are based on cross-validation, for the sake of completeness it is noteworthy to consider the scenarios of real-world applications where the future prediction is based on past data. We therefore design three experiments by considering the reports published in 2013, 2014, and 2015 as test set and the reports published before each year as training set (only 2012, 2012–2013, and 2012–2014 respectively). The results of predicting the reports of each year together with the cross validation scenario (CV) are shown in Figure 2b. While the performance becomes slightly worse in the target years 2013 and 2015, in general the combination of textual and market features can explain approximately half of volatility in the financial system.

## 6.3 Sectors

Corporations in the same sector share not only similar products or services but also risks and instability factors. Considering the sentiment of the financial system as a homogeneous body may neglect the specific factors of each sector. We therefore set out to investigate the existence and nature of these differences.

We start by observing the prediction performance on different sectors: We use our method from the previous section, but split the test set across sectors and plot the results in Figure 2c. The hashed areas indicate the GARCH and Market baselines for the Text and Text+Market feature sets, respectively. We observe considerable differences between the performance of the sectors, especially when using only sentiment analysis methods (i.e. only text features).

Table 3: Number of reports per sectors

| ene | ind | hlth | fin | tech | pub |
|-----|-----|------|-----|------|-----|
| 187 | 160 | 305 | 847 | 408 | 217 |
| n-dur | dur | capt | serv | misc | |
| 151 | 115 | 255 | 639 | 153 | |

Given these differences and also the probable similarities between the risk factors of the reports in the same sector, a question immediately arises: can training different models for different sectors improve the performance of prediction?

To answer it, for each sector, we train a model using only the subset of the reports in that sector and use 5-fold validation to observe performance. We refer to these models as sector-specific in contrast to the general model, trained on all the data. Figures 3a and 3b compare their results: we can see that the sector-specific bars are lower than the general model ones. This is to some extent surprising, as one would expect that domain-specific training would improve the performance of sentiment analysis in text. However, we need to consider the size of the training set. By training on each sector we have reduced the size of our training sets to those reported in Table 3. To verify the effect of the size of training data, we train a sector-agnostic model for each sector. Each sector-agnostic model is trained by random sampling of a training set of the same size as the set available for its sector from all the reports, but evaluated–similar to sector-specific models–on the test set of the sector. Figures 3a and 3b also plot the results of the sector-agnostic models.

The large performance differences between

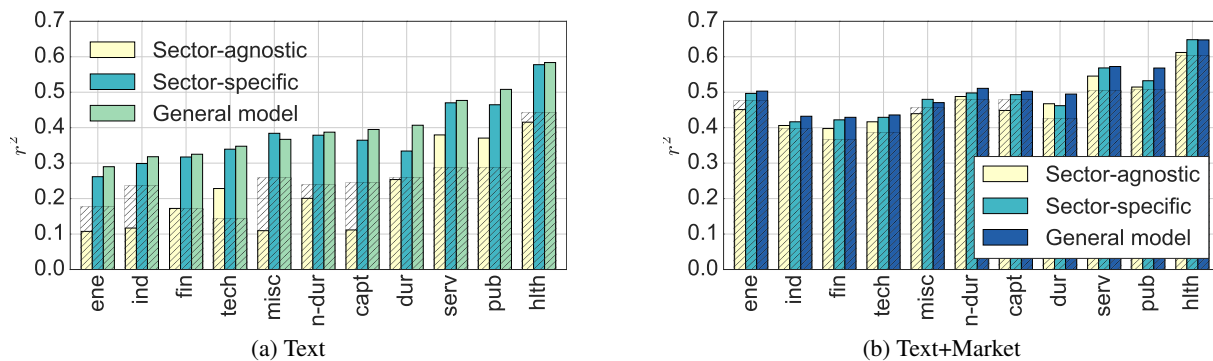

(a) Text

(b) Text+Market

Figure 3: Results when retraining on sector-specific subsets versus the general model and versus subsets of the same size but sector-agnostic. The hashed area in (a) indicates the GARCH and in (b) the Market baseline.

sector-agnostic and -specific show the existence of particular risk factors in each sector and their importance. Results also confirm the hypothesis that the data for training in each sector is simply too small, and as additional data is accumulated, we can further improve on the results by training on different sectors independently.

We continue by examining some examples of essential terms in sectors. To address this, we have to train a linear regression method on all the reports of each sector, without using any dimensionality reduction. Linear regression without dimensionality reduction has the benefit of interpretability: the coefficient of each feature (i.e. term in the lexicon) can be seen as its importance with regards to volatility prediction. After training, we observe that some keywords e.g. *crisis*, or *delist* constantly have high coefficient values in the sector-specific as well as general model. However, some keywords are particularly weighted high in specific-sector models.

For instance, the keyword *fire* has a high coefficient in the energy sector, but very low in the others. The reason is due to the problem of ambiguity i.e. in the energy sector, *fire* is widely used to refer to *explosion* e.g. 'fire and explosion hazards' while in the lexicon, it is stemmed from *firing* and *fired*: the act of dismissing from a job. This later sense of word is however weighted as a low risk-sensitive keyword in the other sectors. Such an ambiguity can indeed be mitigated by sector-specific models since the variety of the words' senses are more restricted inside each sector. Another example is an interesting observation on the word *beneficial*. The word is introduced as a positive sentiment in the lexicon while it gains highly

negative sentiments in some sectors (health care, and basic industries). Investigating in the reports, we observe the broad use of the expression 'beneficial owner' which is normally followed by riskfull sentences since the beneficial owners can potentially influence shareholders' decision power.

## 7 Conclusion

In this work, we studied the sentiment of recent 10-K annual disclosures of companies in stock markets for forecasting volatility. Our bag-of-words sentiment analysis approach benefits from state-of-the-art models in information retrieval which use word embeddings to extend the weight of the terms to the similar terms in the document. Additionally, we explored fusion methods to combine the text features with factual market features, achieved from historical prices i.e. GARCH prediction model, and current volatility. In both cases, our approach outperforms state-of-the-art volatility prediction methods with 10-K reports and demonstrates the effectiveness of sentiment analysis in long-term volatility forecasting.

In addition, we studied the characteristics of each individual sector with regard to risk-sensitive terms. Our analysis shows that reports in same sectors considerably share particular risk and instability factors. However, despite expectations, training different models on different sectors does not improve performance compared to the general model. We traced this to the size of the available data in each sector, and show that there are still benefits in considering sectors, which could be further explored in the future as more data becomes available.

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
