# Peer review of "Volatility Prediction using Financial Disclosures Sentiments with Word Embedding-based IR Models"

_ACL 2017 — decision unknown_

[Official Review · Reviewer 1 · rating 2 · confidence 3]
soundness 5 · originality 5 · clarity 4 · impact 3 · substance 3 · appropriateness 4 · meaningful comparison 3 · presentation format Poster

- Strengths:
The approach described in the manuscript outperformed the previous approaches
and achieved the state-of-the-art result.

Regarding data, the method used the combination of market and text data.

The approach used word embeddings to define the weight of each lexicon term by
extending it to the similar terms in the document.

- Weaknesses:
Deep-learning based methods were known to be able to achieve relatively good
performances without much feature engineering in sentimental analysis. More
literature search is needed to compare with the related works would be better.

The approach generally improved performance by feature-based methods without
much novelty in model or proposal of new features.

- General Discussion:
The manuscript described an approach in sentimental analysis. The method used a
relatively new method of using word embeddings to define the weight of each
lexicon term. However, the novelty is not significant enough.

[Official Review · Reviewer 2 · rating 2 · confidence 4]
soundness 5 · originality 5 · clarity 4 · impact 3 · substance 4 · appropriateness 5 · meaningful comparison 3 · presentation format Poster

- Strengths:

- Weaknesses:

- General Discussion:

This paper investigates sentiment signals in  companies’ annual 10-K filing
reports to forecast volatility. 

The authors evaluate information retrieval term weighting models which are
seeded with a finance-oriented sentiment lexicon and expanded with word
embeddings. PCA is used to reduce dimensionality before Support Vector
Regression is applied for similarity estimation.

In addition to text-based features, the authors also use non-text-based market
features (e.g. sector information and volatility estimates).

Multiple fusion methods to combine text features with market features are
evaluated.

COMMENTS

It would be interesting to include two more experimental conditions, namely 1)
a simple trigram SVM which does not use any prior sentiment lexica, and 2)
features that reflect delta-IDFs scores for individual features.
As an additional baseline, it would be good to see binary features.

This paper could corroborate your references:

https://pdfs.semanticscholar.org/57d6/29615c19caa7ae6e0ef2163eebe3b272e65a.pdf